

# OSoMe: the IUNI observatory on social media

Clayton A. Davis[1,2,*], Giovanni Luca Ciampaglia[1,3,*], Luca Maria Aiello[4], Keychul Chung[2], Michael D. Conover[5], Emilio Ferrara[6], Alessandro Flammini[1,2,3], Geoffrey C. Fox[2], Xiaoming Gao[7], Bruno Gonçalves[8], Przemyslaw A. Grabowicz[9], Kibeom Hong[2], Pik-Mai Hui[1,2], Scott McCaulay[3], Karissa McKelvey[10], Mark R. Meiss[11], Snehal Patil[12], Chathuri Peli Kankanamalage[3], Valentin Pentchev[3], Judy Qiu[2], Jacob Ratkiewicz[11], Alex Rudnick[11], Benjamin Serrette[3], Prashant Shiralkar[1,2], Onur Varol[1,2], Lilian Weng[13], Tak-Lon Wu[14], Andrew J. Younge[2] and Filippo Menczer[1,2,3]

[1] Center for Complex Networks and Systems Research, Indiana University, Bloomington, United States
[2] School of Informatics and Computing, Indiana University, Bloomington, United States
[3] Network Science Institute, Indiana University, Bloomington, United States
[4] Bell Labs, London, United Kingdom
[5] LinkedIn Inc., Mountain View, CA, United States
[6] Information Sciences Institute, University of Southern California, Marina Del Rey, CA, United States
[7] Facebook Inc., Boston, MA, United States
[8] Center for Data Science, New York University, New York, NY, United States
[9] Max Planck Institute for Software Systems Saarbrücken, Germany,
[10] US Open Data, Oakland, CA, United States
[11] Google Inc., Mountain View, CA, United States
[12] Yahoo Inc., Sunnyvale, CA, United States
[13] Affirm Inc., San Francisco, CA, United States
[14] Amazon Inc., Seattle, WA, United States
[*] These authors contributed equally to this work.

Corresponding author
Giovanni Luca Ciampaglia,
gciampag@indiana.edu

## ABSTRACT

The study of social phenomena is becoming increasingly reliant on big data from online social networks. Broad access to social media data, however, requires software development skills that not all researchers possess. Here we present the *IUNI Observatory on Social Media*, an open analytics platform designed to facilitate computational social science. The system leverages a historical, ongoing collection of over 70 billion public messages from Twitter. We illustrate a number of interactive open-source tools to retrieve, visualize, and analyze derived data from this collection. The Observatory, now available at osome.iuni.iu.edu, is the result of a large, six-year collaborative effort coordinated by the Indiana University Network Science Institute.

## INTRODUCTION

The collective processes of production, consumption, and diffusion of information on social media are starting to reveal a significant portion of human social life, yet scientists

struggle to get access to data about it. Recent research has shown that social media can perform as 'sensors' for collective activity at multiple scales (*Lazer et al., 2009*). As a consequence, data extracted from social media platforms are increasingly used side-by-side with—and sometimes even replacing—traditional methods to investigate hard-pressing questions in the social, behavioral, and economic (SBE) sciences (*King, 2011*; *Moran et al., 2014*; *Einav & Levin, 2014*). For example, interpersonal connections from Facebook have been used to replicate the famous experiment by *Travers & Milgram (1969)* on a global scale (*Backstrom et al., 2012*); the emotional content of social media streams has been used to estimate macroeconomic quantities in country-wide economies (*Bollen, Mao & Zeng, 2011*; *Choi & Varian, 2012*; *Antenucci et al., 2014*); and imagery from Instagram has been mined (*De Choudhury et al., 2013*; *Andalibi, Ozturk & Forte, 2015*) to understand the spread of depression among teenagers (*Link et al., 1999*).

A significant amount of work about information production, consumption, and diffusion has been thus aimed at modeling these processes and empirically discriminating among models of mechanisms driving the spread of memes on social media networks such as Twitter (*Guille et al., 2013*). A set of research questions relate to how social network structure, user interests, competition for finite attention, and other factors affect the manner in which information is disseminated and why some ideas cause viral explosions while others are quickly forgotten. Such questions have been addressed both in an empirical and in more theoretical terms.

Examples of empirical works concerned with these questions include geographic and temporal patterns in social movements (*Conover et al., 2013b*; *Conover et al., 2013a*; *Varol et al., 2014*), the polarization of online political discourse (*Conover et al., 2011b*; *Conover et al., 2011a*; *Conover et al., 2012*), the use of social media data to predict election outcomes (*DiGrazia et al., 2013*) and stock market movements (*Bollen, Mao & Zeng, 2011*), the geographic diffusion of trending topics (*Ferrara et al., 2013*), and the lifecycle of information in the attention economy (*Ciampaglia, Flammini & Menczer, 2015*).

On the more theoretical side, agent-based models have been proposed to explain how limited individual attention affects what information we propagate (*Weng et al., 2012*), what social connections we make (*Weng et al., 2013*), and how the structure of social and topical networks can help predict which memes are likely to become viral (*Weng, Menczer & Ahn, 2013*; *Weng, Menczer & Ahn, 2014*; *Nematzadeh et al., 2014*; *Weng & Menczer, 2015*).

Broad access by the research community to social media platforms is, however, limited by a host of factors. One obvious limitation is due to the commercial nature of these services. On these platforms, data are collected as part of normal operations, but this is seldom done keeping in mind the needs of researchers. In some cases researchers have been allowed to harvest data through programmatic interfaces, or APIs. However, the information that a single researcher can gather through an API typically offers only a limited view of the phenomena under study; access to historical data is often restricted or unavailable (*Zimmer, 2015*). Moreover, these samples are often collected using ad-hoc procedures, and the statistical biases introduced by these practices are only starting to be understood (*Morstatter et al., 2013*; *Ruths & Pfeffer, 2014*; *Hargittai, 2015*).

A second limitation is related to the ease of use of APIs, which are usually meant for software developers, not researchers. While researchers in the SBE sciences are increasingly acquiring software development skills (*Terna, 1998*; *Raento, Oulasvirta & Eagle, 2009*; *Healy & Moody, 2014*), and intuitive user interfaces are becoming more ubiquitous, many tasks remain challenging enough to hinder research advances. This is especially true for those tasks related to the application of fast visualization techniques.

A third, important limitation is related to user privacy. Unfettered access to sensitive, private data about the choices, behaviors, and preferences of individuals is happening at an increasing rate (*Tene & Polonetsky, 2012*). Coupled with the possibility to manipulate the environment presented to users (*Kramer, Guillory & Hancock, 2014*), this has raised in more than one occasion deep ethical concerns in both the public and the scientific community (*Kahn, Vayena & Mastroianni, 2014*; *Fiske & Hauser, 2014*; *Harriman & Patel, 2014*; *Vayena et al., 2015*).

These limitations point to a critical need for opening social media platforms to researchers in ways that are both respectful of user privacy requirements and aware of the needs of SBE researchers. In the absence of such systems, SBE researchers will have to increasingly rely on closed or opaque data sources, making it more difficult to reproduce and replicate findings—a practice of increasing concern given recent findings about replicability in the SBE sciences (*Open Science Collaboration, 2015*).

Our long-term goal is to enable SBE researchers and the general public to openly access relevant social media data. As a concrete milestone of our project, here we present an *Observatory on Social Media*—an open infrastructure for sharing public data about information that is spread and collected through online social networks. Our initial focus has been on Twitter as a source of public microblogging posts. The infrastructure takes care of storing, indexing, and analyzing public collections and historical archives of big social data; it does so in an easy-to-use way, enabling broad access from scientists and other stakeholders, like journalists and the general public. We envision that data and analytics from social media will be integrated within a nation-wide network of social observatories. These data centers would allow access to a broad range of data about social, behavioral, and economic phenomena nationwide (*King, 2011*; *Moran et al., 2014*; *Difranzo et al., 2014*).

Our team has been working toward this vision since 2010, when we started collecting public tweets to visualize, analyze, and model meme diffusion networks. The IUNI Observatory on Social Media (OSoMe) presented here was launched in early May 2016. It was developed through a collaboration between the Indiana University Network Science Institute (IUNI, iuni.iu.edu), the IU School of Informatics and Computing (SoIC, soic.indiana.edu), and the Center for Complex Networks and Systems Research (CNetS, cnets.indiana.edu). It is available at osome.iuni.iu.edu.[1]

## DATA SOURCE

Social media data possess unique characteristics. Besides rich textual content, explicit information about the originating social context is generally available. Information often includes timestamps, geolocations, and interpersonal ties. The Twitter dataset is

[1] The OSoMe website is also available at truthy.indiana.edu. The original website was created to host our first demo, motivated by the application of social media analytics to the study of "astroturf," or artificial grassroots social media campaigns orchestrated through fake accounts and social bots (*Ratkiewicz et al., 2011b*). The *Truthy* nickname was later adopted in the media to refer to the entire project. The current website includes information about other research projects on information diffusion and social bots from our lab.

[2] Research based on this data was deemed exempt from review by the Indiana University IRB under Protocol # 1102004860.

a prototypical example (*McKelvey & Menczer, 2013b*; *McKelvey & Menczer, 2013a*). The Observatory on Social Media is built around a Terabyte-scale historical (and ongoing) collection of tweets. The data source is a large sample of public posts made available by Twitter through elevated access to their streaming API, granted to a number of academic labs. All of the tweets from this sample are stored, resulting in a corpus of approximately **70 billion public tweets** dating back to mid-2010.[2]

An important caveat about the use of these data for research is that possible sampling biases are unknown. When Twitter first made this stream available to the research community, it indicated that the stream contains a random 10% sample of all public tweets. However, no further information about the sampling method was disclosed. Other streaming APIs have been shown to provide a non-uniform sample (*Morstatter et al., 2013*). Even assuming that tweets are randomly sampled, it should be noted that the collection does not automatically translate into a representative sample of the underlying population of Twitter users, or of the topics discussed. This is because the distribution of activity is highly skewed across users and topics (*Weng et al., 2012*) and, as a result, active users and popular topics are better represented in the sample. Sampling biases may also have evolved over time. For example, the fraction of public tweets with exact geolocation coordinates has decreased from approximately 3% in the past to approximately 0.3% due to the recent change of location privacy settings in Twitter mobile clients from "opt-out" to "opt-in." This change was motivated by public privacy concerns about location tracking. This and other platform changes may significantly affect the composition of our sample in ways that we are unable to assess.

The high-speed stream from which the data originates has a rate that ranges in the order of $10^6 - 10^8$ tweets/day. Figure 1 illustrates the growth of the Twitter collection over time.

## SYSTEM ARCHITECTURE

Performing analytics at this scale presents specific challenges. The most obvious has to do with the design of a suitable architecture for processing such a large volume of data. This requires a scalable storage substrate and efficient query mechanisms.

The core of our system is based on a distributed storage cluster composed of 10 compute nodes, each with $12 \times$ 3TB disk drives, $2 \times$ 146 GB RAID-1 drives for the operative system, 64 GB RAM, and $2\times$ Xeon 2650 CPUs with 8 cores each (16 total per node). Access to the nodes is provided via two head nodes, each equipped with 64 GB RAM, and $4\times$ Xeon 2650 CPUs with four cores each (24 total per node), using 1GB ethernet infiniband.

The software architecture the Observatory builds upon the Apache Big Data Stack (ABDS) framework (*Jha et al., 2014*; *Qiu et al., 2014*; *Fox et al., 2014*). Development has been driven over the years by the need for increasingly demanding social media analytics applications (*Gao, Nachankar & Qiu, 2011*; *Gao & Qiu, 2013*; *Gao & Qiu, 2014*; *Gao et al., 2014*; *Gao, Ferrara & Qiu, 2015*; *Wu et al., in press*). A key idea behind our enhancement of the ABDS architecture is the shift from standalone systems to modules; multiple modules can be used within existing software ecosystems. In particular, we have focused our efforts on enhancing two well-known Apache modules, Hadoop (*The Apache Software Foundation, 2016b*) and HBase (*The Apache Software Foundation, 2016a*).

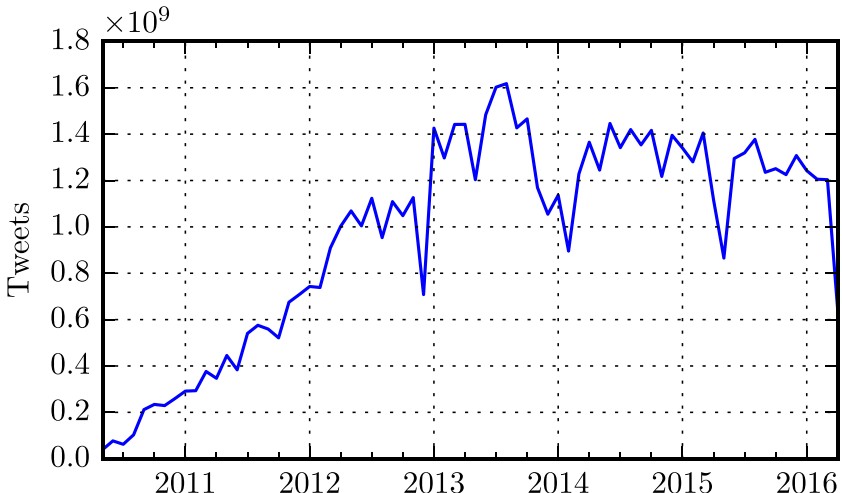

**Figure 1** **Number of monthly messages collected and indexed by OSoME.** System failures have caused occasional interruptions of the collection system.

The architecture is illustrated in Fig. 2. The *data collection system* receives data from the Twitter Streaming API. Data are first stored on a temporary location and then loaded into the distributed storage layer on a daily basis. At the same time, *long-term backups* are stored on tape to allow recovery in case of data loss or catastrophic events.

The design of the *NoSQL distributed DB* module was guided by the observation that queries of social media data often involve unique constraints on the textual and social context such as temporal or network information. To address this issue, we leveraged the HBase system as the storage substrate and extended it with a flexible indexing framework. The resulting *IndexedHBase* module allows one to define fully customizable text index structures that are not supported by current state-of-the-art text indexing systems, such as Solr (*The Apache Software Foundation, 2016c*). The custom index structures can embed contextual information necessary for efficient query evaluation. The IndexedHBase software is publicly available (*Wiggins, Gao & Qiu, 2016*).

The pipelines commonly used for social media data analysis consist of multiple algorithms with varying computation and communication patterns. For example, building the network of retweets of a given hashtag will take more time and computational resources than just counting the number of posts containing the hashtag. Moreover, the temporal resolution and aggregation windows of the data could vary dramatically, from seconds to years. A number of different processing frameworks could be needed to perform such a wide range of tasks. To design the *analytics* module of the Observatory we choose Hadoop, a standard framework for Big Data analytics. We use YARN (*The Apache Software Foundation, 2016d*) to achieve efficient execution of the whole pipeline, and integrate it with IndexedHBase. An advantage deriving from this choice is that the overall software stack can dynamically adopt different processing frameworks to complete heterogeneous tasks of variable size.

A distributed task queue, and an in-memory key/value store implement the *middleware* layer needed to submit queries to the backend of the Observatory. We use Celery (*Solem &*

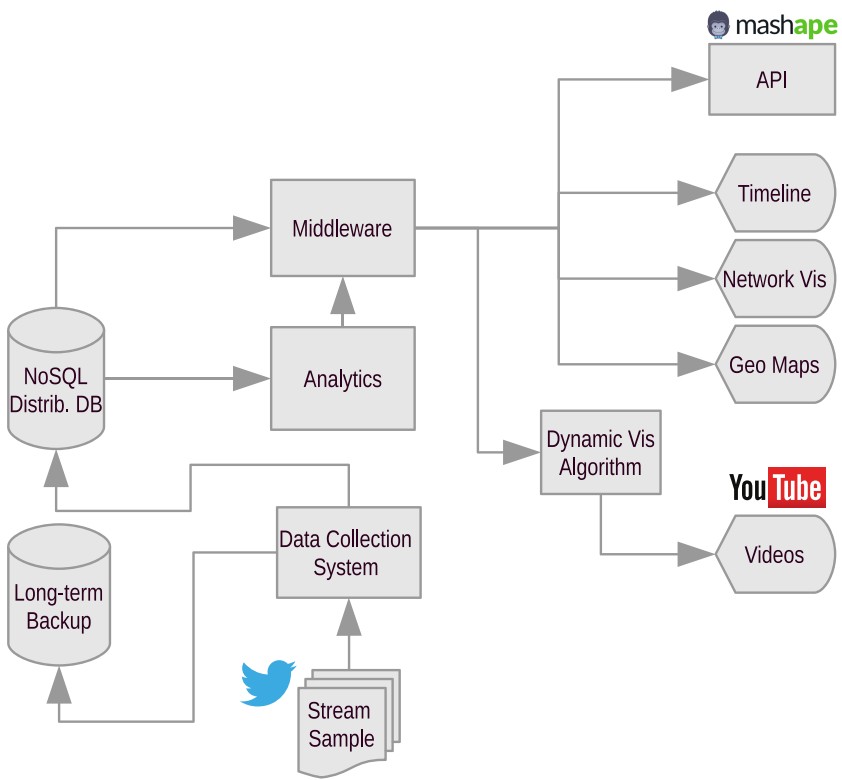

**Figure 2** **Flowchart diagram of the OSoMe architecture.** Arrows indicate flow of data.

*Contributors, 2016*) and Redis (*Sanfilippo, 2016*) to implement such layer. The task queue limits the number of concurrent jobs processed by the system according to the task type (index-only vs. map/reduce) to prevent extreme degradation of performance due to very high load.

The Observatory user interface follows a modular architecture too, and is based on a number of apps, which we describe in greater detail in the following section. Three of the apps (*Timeline*, *Network visualization*, and *Geographic maps*) are directly accessible within OSoMe through Web interfaces. We rely on the popular video-sharing service YouTube for the fourth app, which generates meme diffusion movies (*Videos*). The movies are rendered using a fast *dynamic visualization algorithm* that we specifically designed for temporal networks. The algorithm captures only the most persistent trends in the temporal evolution, at the expense of high-frequency churn (*Grabowicz, Aiello & Menczer, 2014*). The software is freely available (*Grabowicz & Aiello, 2013*). Finally, the Observatory provides access to raw data via a programmatic interface (*API*).

## APPLICATIONS

Storing and indexing tens of billions of tweets is of course pointless without a way to make sense of such a huge trove of information. The Observatory lowers the barrier of entry to social media analysis by providing users with several ready-to-use, Web-based data visualization tools. Visualization techniques allow users to make sense of complex data

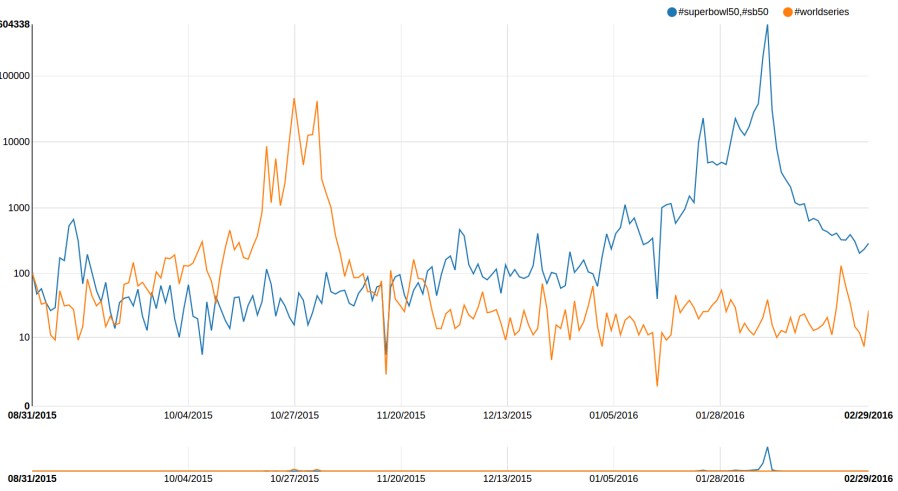

**Figure 3  Number of tweets per day about the Super Bowl (in blue) and the World Series (in orange), from September 2015 through February 2016.** The $Y$-axis is in logarithmic scale, shifted by one to account for null counts. The plot shows two outages in the data collection that occurred around mid-November 2015 and mid-January 2016.

and patterns (*Card, 2009*), and let them explore the data and try different visualization parameters (*Rafaeli, 1988*). In the following, we give a brief overview of the available tools.

It is important to note that, in compliance with the Twitter terms of service (*Twitter, Inc., 2016*), OSoMe does not provide access to the content of tweets, nor of Twitter user objects. However, researchers can obtain numeric object identifiers of tweets in response to their queries. This information can then be used to retrieve tweet content via the official Twitter API. (There is one exception, described below.) Another necessary step to comply with the Twitter terms is to remove deleted tweets from the database. Using a Redis queue, we collect deletion notices from the public Twitter stream, and then feed them to a backend task for deletion.

## Temporal Trends

The *Trends* tool produces time series plots of the number of tweets including one or more given hashtags; it can be compared to the service provided by Google Trends, which allows users to examine the interest toward a topic reflected by the volume of search queries submitted to Google over time.

Users may specify multiple terms in one query, in which case all tweets containing any of the terms will be computed; and they can perform multiple queries, to allow comparisons between different topics. For example, let us compare the relative tweet volumes about the World Series and the Superbowl. We want our Super Bowl timeline to count tweets containing any of #SuperBowl, #SuperBowl50, or #SB50. Since hashtags are case-insensitive and we allow trailing wildcards, this query would be " #superbowl*, #sb50." Adding a timeline for the " #worldseries" query results in the plot seen in Fig. 3. Each query on the Trends tool takes 5–10 s; this makes the tool especially suitable for interactive exploration of Twitter conversation topics.

## Diffusion and co-occurrence networks

In a diffusion network, nodes represent users and an edge drawn between any two nodes indicates an exchange of information between those two users. For example, a user could rebroadcast (*retweet*) the status of another user to her followers, or she could address another user in one of her statuses by including a mention to their user handle (*mention*). Edges have a weight to represent the number of messages connecting two nodes. They may also have an intrinsic direction to represent the flow of information. For example, in the retweet network for the hashtag #IceBucketChallenge, an edge from user $i$ to user $j$ indicates that $j$ retweeted tweets by $i$ containing the hashtag #IceBucketChallenge. Similarly, in a mention network, an edge from $i$ to $j$ indicates that $i$ mentioned $j$ in tweets containing the hashtag. Information diffusion network, sometimes also called information cascades, have been the subject of intense study in recent years (*Gruhl et al., 2004*; *Weng et al., 2012*; *Bakshy et al., 2012*; *Weng et al., 2013*; *Weng, Menczer & Ahn, 2013*; *Romero, Meeder & Kleinberg, 2011*).

Another type of network visualizes how hashtags co-occur with each other. Co-occurrence networks are also weighted, but undirected: nodes represent hashtags, and the weight of an edge between two nodes is the number of tweets containing both of those hashtags.

OSoMe provides two tools that allow users to explore diffusion and and co-occurrence networks.

### Interactive network visualization

The *Networks* tool enables the visualization of how a given hashtag spreads through the social network via retweets and mentions (Fig. 4) or what hashtags co-occur with a given hashtag. The resulting network diagrams, created using a force-directed layout (*Kamada & Kawai, 1989*), can reveal topological patterns such as influential or highly-connected users and tightly-knit communities. Users can click on the nodes and edges to find out more information about the entities displayed—users, tweets, retweets, and mentions—directly from Twitter. Network are cached to enable fast access to previously-created visualizations.

For visualization purposes, the size of large networks is reduced by extracting their $k$-core (*Alvarez-Hamelin et al., 2005*) with $k$ sufficiently large to display 1,000 nodes or less ($k = 5$ in the example of Fig. 4). The use of this type of filter implies a bias toward densely connected portions of diffusion networks, where most of the activity occurs, and toward the most influential participants.

The tool allows access to Twitter content, such as hashtags and user screen names. This content is available both through the interactive interface itself, and as a downloadable JSON file. To comply with the Twitter terms, the $k$-core filter also limits the number of edges (tweets).

### Animations

Because tweet data are time resolved, the evolution of a diffusion or co-occurrence network can be also visualized over time. Currently the *Networks* tool visualizes only static networks aggregated over the entire search period specified by the user; we aim to add the ability to observe the network evolution over time, but in the meantime we also provide the *Movies*

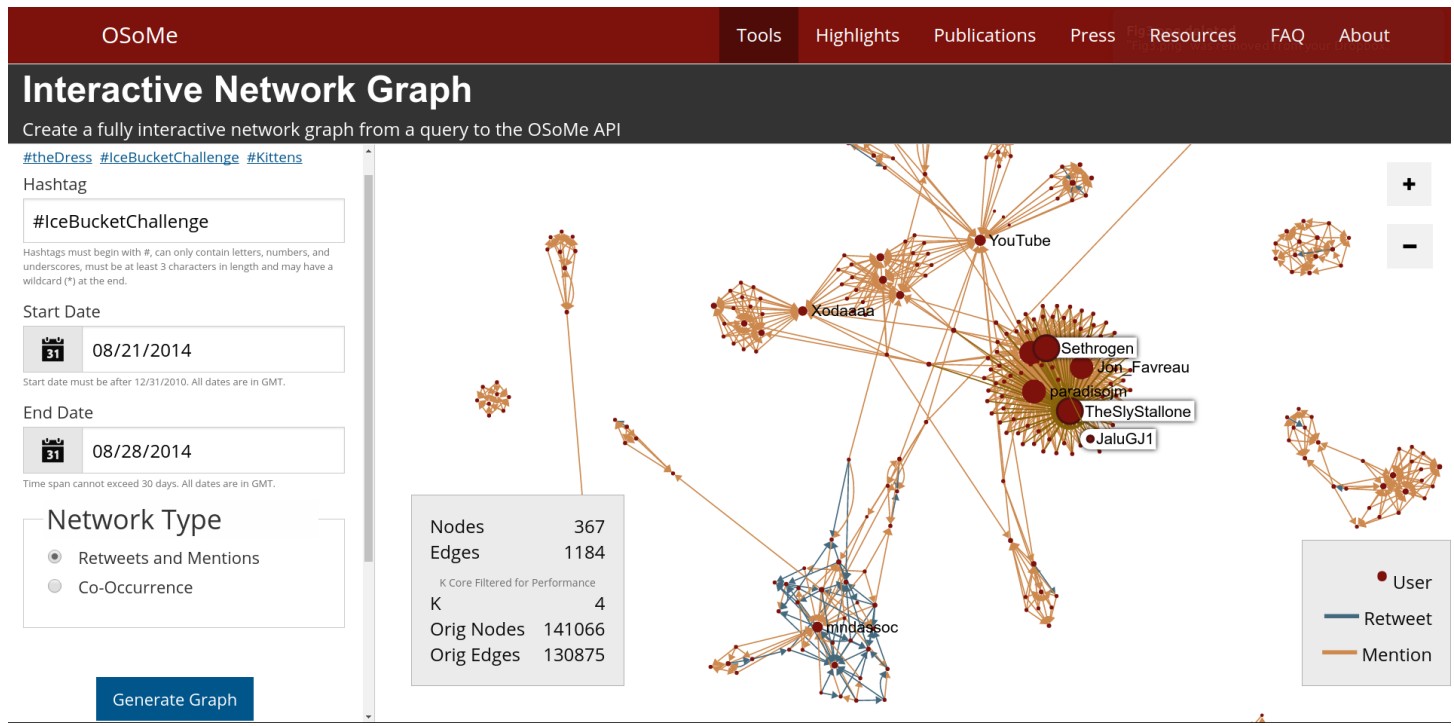

**Figure 4   Interactive network visualization tool.** A detail of the network of retweets and mention for a hashtag commonly linked to "Ice Bucket Challenge," a popular Internet phenomenon from 2014. The size of a node is proportional to its strength (weighted degree). The detail shows the patterns of mention and information broadcasting occurring between celebrities, as the viral challenge was taking off.

tool, an alternative service that lets users generate animations of such processes (Fig. 5). We have successfully experimented with fast visualization techniques in the past, and have found that edge filtering is the best approach for efficiently visualizing networks that undergo a rapid churn of both edges and nodes. We have therefore deployed a fast filtering algorithm developed by our team (*Grabowicz, Aiello & Menczer, 2014*). The user-generated videos are uploaded to YouTube, and we cache the videos in case multiple users try to visualize the same network.

## Geographic maps

Online social networks are implicitly embedded in space, and the spatial patterns of information spread have started to be investigated in recent years (*Ferrara et al., 2013*; *Conover et al., 2013a*). The *Maps* tool enables the exploration of information diffusion through geographic space and time. A subset of tweets contain exact latitude/longitude coordinates in their metadata. By aggregating these coordinates into a heatmap layer superimposed on a world map, one can observe the geographic signature of the attention being paid to a given meme. Figure 6 shows an example. Our online tool goes one step further, allowing the user to explore how this geographic signature evolves over a specified time period, via a slider widget.

It takes at least 30 s to prepare one of these visualizations *ex novo*. We hope to reduce this lead time with some backend indexing improvements. To enable exploration, we cache all created heatmaps for a period of one week. While cached, the heatmaps can be

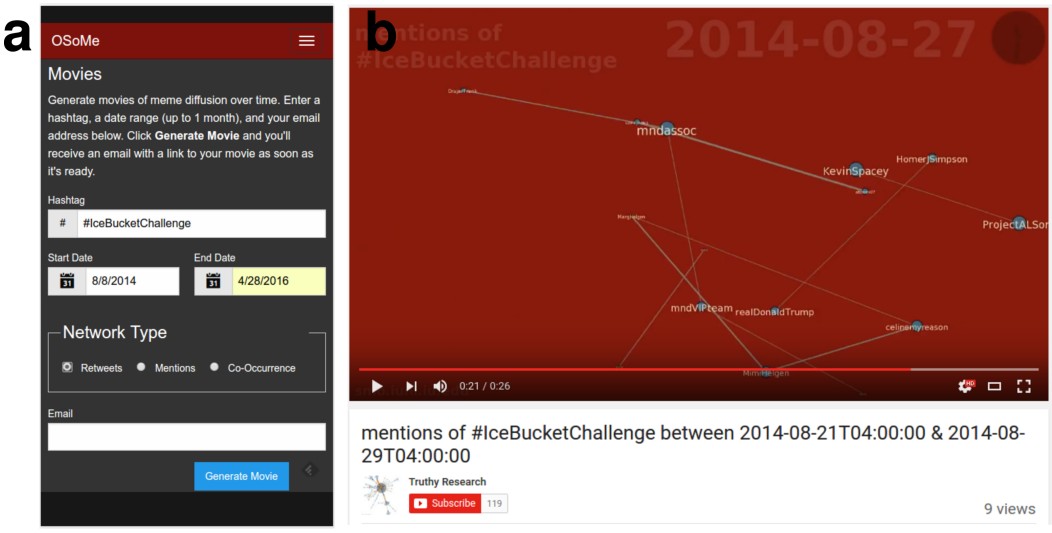

**Figure 5** **Temporal information diffusion movies.** (A) The interface of the *Movies* tool let users specify a hashtag, a temporal interval, and the type of diffusion ties to visualize (retweets, mentions, or hashtag co-occurrence). (B) Example of a generated movie frame, showing a retweet network for the #IceBucketChallenge hashtag. YouTube: https://www.youtube.com/watch?v=nZkHRtPciYU.

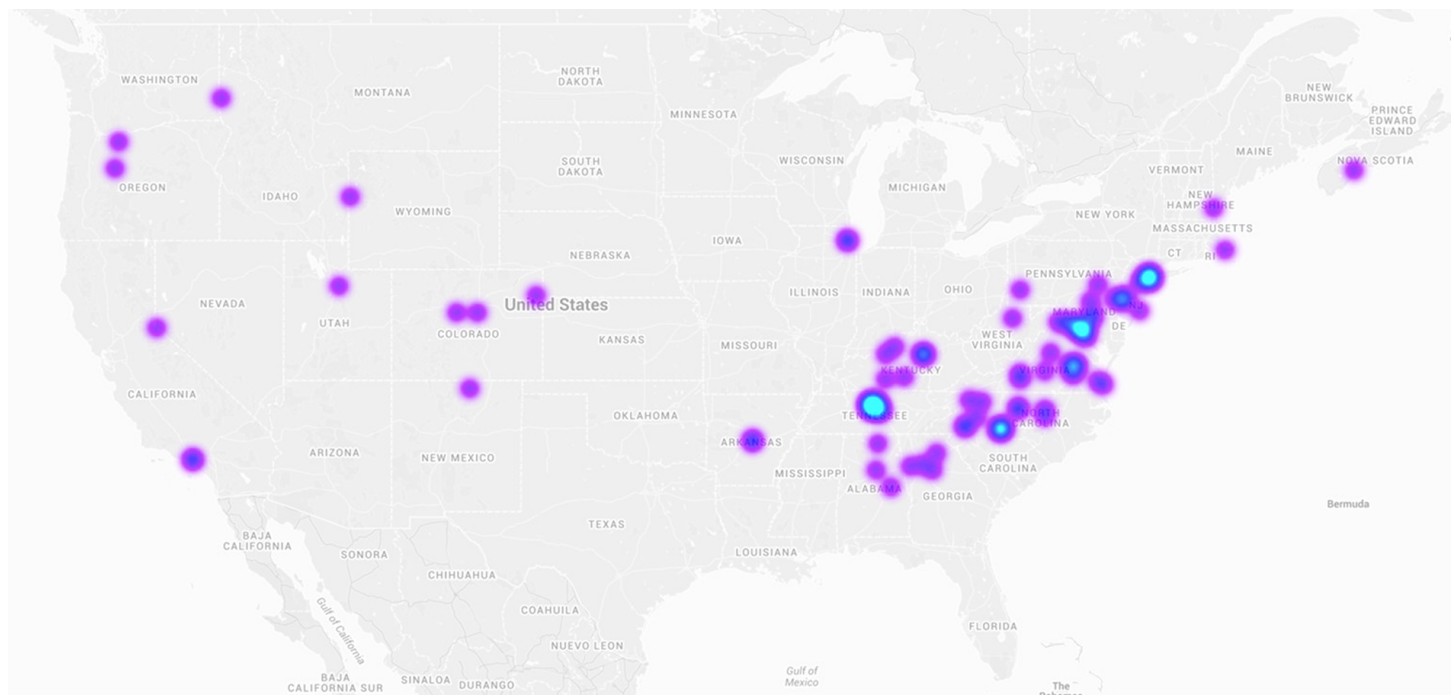

**Figure 6** **Heatmap of tweets containing the hashtag #snow on January 22, 2016, the day of a large snowstorm over the Eastern United States.**

retrieved instantly, enabling other users to browse and interact with these previously-created visualizations. In the future we hope to experiment with overlaying diffusion networks on top of geographical maps, for example using multi-scale backbone extraction (*Serrano, Boguná & Vespignani, 2009*) and edge bundling techniques (*Selassie, Heller & Heer, 2011*).

An important caveat for the use of the maps tool is that it is based on the very small percentage of tweets that contain exact geolocation coordinates. Furthermore, as already discussed, this percentage has changed over time.

## API

We expect that the majority of users of the Observatory will interact with its data primarily through the tools described above. However, since more advanced data needs are to be expected, we also provide a way to export the data for those who wish to create their own visualizations and develop custom analyses. This is possible either within the tools, via export buttons, and through a read-only HTTP API.

The OSoMe API is deployed via the Mashape management service. Four public methods are currently available. Each takes as input a time interval and a list of tokens (hashtags and/or usernames):

- `tweet-id`: returns a list of tweet IDs mentioning at least one of the inputs in the given interval;
- `counts`: returns a count of the number of tweets mentioning each input token in the given interval;
- `time-series`: for each day in the given time interval, returns a count of tweets matching any of the input tokens;
- `user-post-count`: returns a list of user IDs mentioning any of the tokens in the given time frame, along with a count of matching tweets produced by each user.

## EVALUATION

In the first several weeks since launch, the OSoMe infrastructure has served a large number of requests, as shown in Fig. 7. The spike corresponds to May 6, 2016, the date of a press release about the launch. Most of these requests complete successfully, with no particular deterioration for increasing loads (Fig. 8).

To evaluate the scalability of the Hadoop-based analytics tools with increasing data size, we plot in Fig. 9 the run time of queries submitted by users through OSoMe interactive tools, as a function of the number of tweets matching the query parameters. We observe a sublinear growth, suggesting that the system scales well with job size. A job may take from approximately 30 s to several hours depending on many factors such as system load and number of tweets processed. However, even different queries that process the same number of tweets may perform differently, depending on the width of the query time window. This is partly due to "hotspotting": the temporal locality of our data layout across the nodes of the storage cluster causes decreases in performance when different Hadoop mappers access the same disks. A query spanning a short period of time runs slower than one matching the same number of tweets over a longer period. These results suggest that our data layout

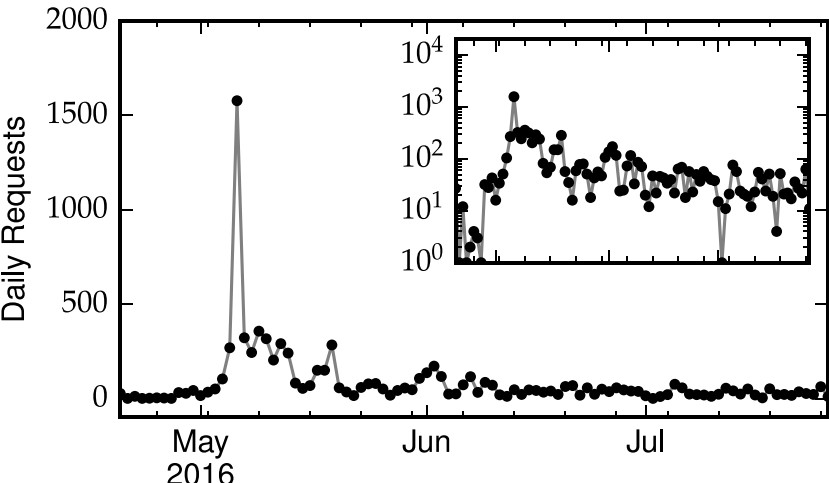

**Figure 7 Number of daily requests to the Observatory, including both API calls and interactive queries.** The inset shows the same data, on a logarithmic scale.

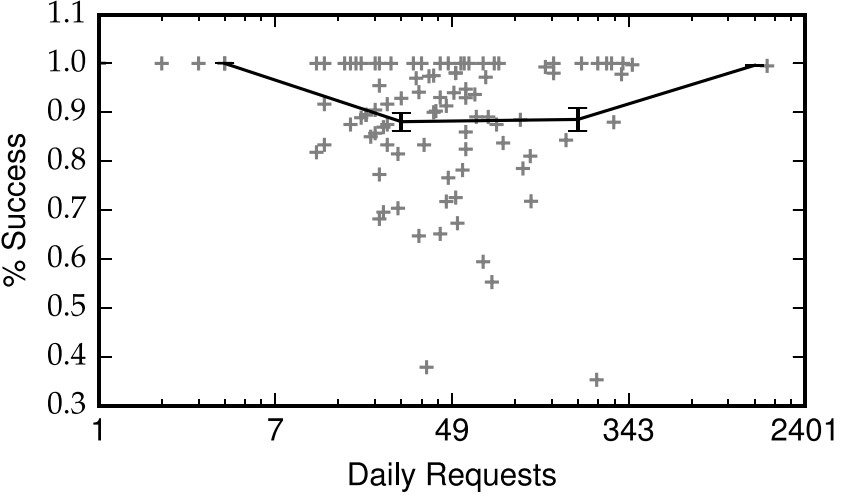

**Figure 8 Fraction of successful requests as a function of daily request load.** Error bars are standard errors within logarithmic bins.

design may need to be reconsidered in future development. An alternative approach to improve performance of queries is to entirely remove the bottleneck of Hadoop processing by indexing additional data. For example, in the Networks and Movies tools, we could index the retweet and mention edges. The resulting queries would utilize the indices only, resulting in response times comparable to those of the Trends tool.

Finally, we tested the scalability of queries using the HBase index with the load. Figure 10 shows that the total run time is not strongly affected by the number of concurrent jobs, up to the size of the task queue (32). For larger loads, run time scales linearly as expected.

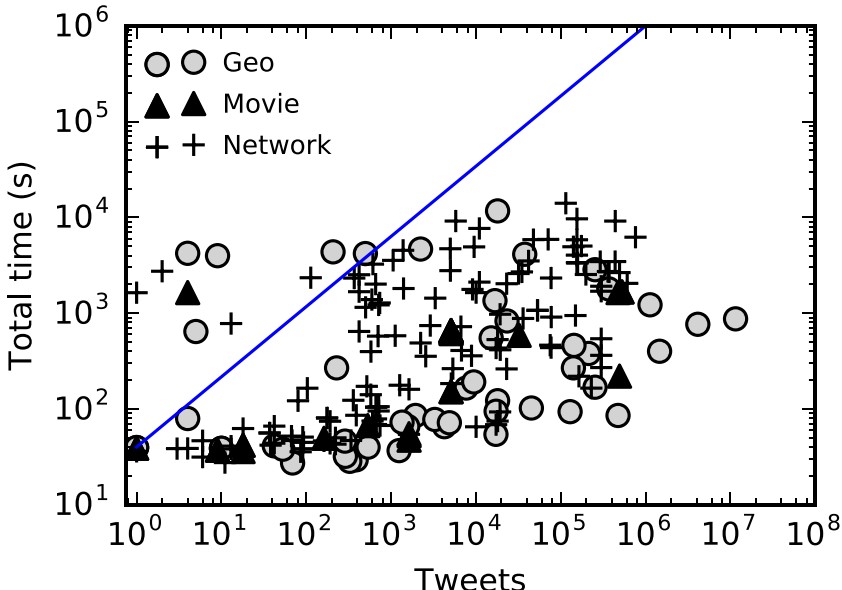

**Figure 9  Run time versus number of tweets processed by Hadoop-based interactive tools.** The line is a guide for the eye corresponding to linear growth.

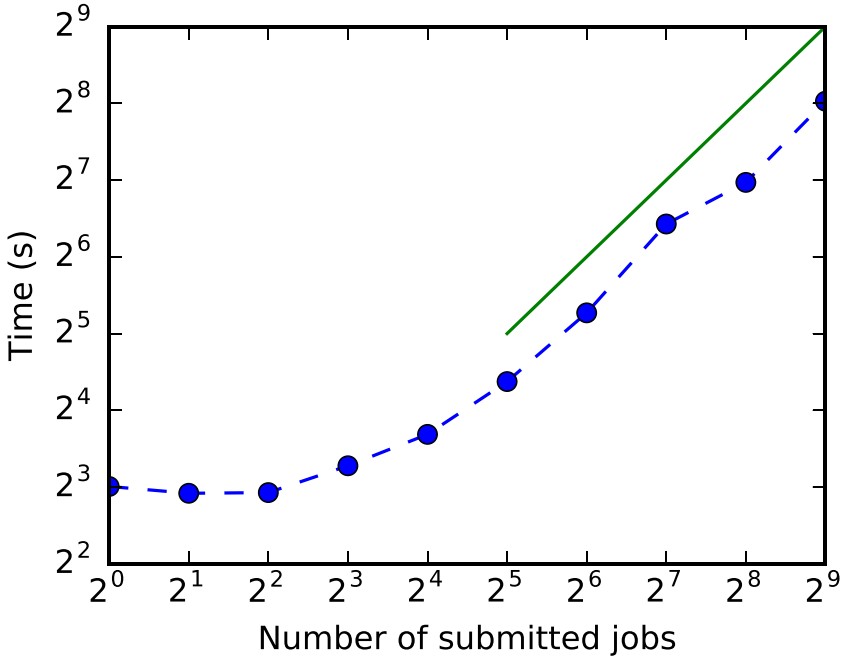

**Figure 10  Run time versus number of concurrent jobs that use the HBase index.** The line is a guide for the eye representing linear growth.

## CONCLUSION

The IUNI Observatory on Social Media is the culmination of a large collaborative effort at Indiana University that took place over the course of six years. We hope that it will facilitate computational social science and make big social data easier to analyze by a broad community of researchers, reporters, and the general public. The lessons learned during the development of the infrastructure may be helpful for future endeavors to foster data-intensive research in the social, behavioral, and economic sciences.

We welcome feedback from researchers and other end users about usability and usefulness of the tools presented here. In the future, we plan to carry out user studies and tutorial workshops to gain feedback on effectiveness of the user interfaces, efficiency of the tools, and desirable extensions.

We encourage the research community to create new social media analytic tools by building upon our system. As an illustration, we created a mashup of the OSoMe API with the BotOrNot API (*Davis et al., 2016*), also developed by our team, to evaluate the extent to which Twitter campaigns are sustained by social bots. The software is freely available online (*Davis, 2016*).

The opportunities that arise from the Observatory, and from computational social science in general, could have broad societal impact. Systematic attempts to mislead the public on a large scale through "astroturf" campaigns and social bots have been uncovered using big social data analytics, inspiring the development of machine learning methods to detect these abuses (*Ratkiewicz et al., 2011a*; *Ferrara et al., 2016*; *Subrahmanian et al., 2016*). Allowing citizens to observe how memes spread online may help raise public awareness of the potential dangers of social media manipulation.

## ACKNOWLEDGEMENTS

The authors would like to acknowledge Alessandro Vespignani and Johan Bollen for discussions leading to the early vision of an Observatory on Social Media; and Rob Henderson, Shing-Shong (Bruce) Shei, Gary Miksik, Allan Streib, and Koji Tanaka for their kind assistance with system administration. We are deeply grateful to Twitter for supporting computational social science research, including the efforts described in this paper, by granting our lab elevated access to the public stream of tweets.

Any opinions, findings, and conclusions or recommendations expressed in this material are those of the author(s) and do not necessarily reflect the views of the funding agencies.

### Funding

This work was supported in part by NSF (grants CCF-1101743 and OCI-1149432), the J.S. McDonnell Foundation (grant 220020274), the Swiss National Science Foundation (fellowship PBTIP2_142353), the Lilly Endowment, the Center for Complex Networks and Systems Research (CNetS), the Digital Science Center (DSC), and the Indiana University

Network Science Institute (IUNI). The funders had no role in study design, data collection and analysis, decision to publish, or preparation of the manuscript.

## Grant Disclosures

The following grant information was disclosed by the authors:
NSF: CCF-1101743, OCI-1149432.
J.S. McDonnell Foundation: PBTIP2_142353.
Lilly Endowment, the Center for Complex Networks and Systems Research.
Digital Science Center.
Indiana University Network Science Institute.

## Competing Interests

Filippo Menczer is an Academic Editor for PeerJ Computer Science. Xiaoming Gao is an employee of Facebook; Luca Maria Aiello is an employee of Bell Labs; Snehal Patil is an employee of Yahoo!; Mike Conover is an employee of LinkedIn; Mark Meiss, Jacob Ratkiewicz, and Alex Rudnick are employees of Google; Lilian Weng is an employee of Affirm; and Tak-Lon Wu is an employee of Amazon.

## Author Contributions

- Clayton A. Davis and Giovanni Luca Ciampaglia wrote the paper, prepared figures and/or tables, performed the computation work, reviewed drafts of the paper.
- Luca Maria Aiello, Keychul Chung, Michael D. Conover, Emilio Ferrara, Xiaoming Gao, Bruno Gonçalves, Przemyslaw A. Grabowicz, Kibeom Hong, Pik-Mai Hui, Scott McCaulay, Karissa McKelvey, Mark R. Meiss, Snehal Patil, Chathuri Peli Kankanamalage, Jacob Ratkiewicz, Alex Rudnick, Prashant Shiralkar, Onur Varol, Lilian Weng, Tak-Lon Wu and Andrew J. Younge performed the computation work, reviewed drafts of the paper.
- Alessandro Flammini, Geoffrey C. Fox, Valentin Pentchev and Judy Qiu reviewed drafts of the paper.
- Benjamin Serrette prepared figures and/or tables, performed the computation work, reviewed drafts of the paper.
- Filippo Menczer wrote the paper, prepared figures and/or tables, reviewed drafts of the paper.

## Ethics

The following information was supplied relating to ethical approvals (i.e., approving body and any reference numbers):

This study was deemed exempt from review by the Indiana University IRB office under Protocol #1102004860.

## Data Availability

Data from the OSoMe are available through an API and through interactive apps. Use of the data is subject to the terms of the Twitter Developer Agreement

(https://dev.twitter.com/overview/terms/agreement). For more information please visit: http://osome.iuni.iu.edu/.

It is important to note that, in compliance with the Twitter terms of service (https://dev.twitter.com/overview/terms/policy), OSoMe does not provide access to the content of tweets. However, researchers can obtain numeric object identifiers in response to their queries. This information can then be used to retrieve tweet content via the official Twitter API.

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
