# Peer review of "OSoMe: the IUNI observatory on social media"

_PeerJ Computer Science, doi:10.7717/peerj-cs.87_

## Round 0.1 · original submission · Major Revisions

· Academic Editor

Major Revisions

First of all, I would like to thank all the reviewers, who have done a fantastic job in reading and commenting the paper. All the comments are very useful for improving the authors' work considerably, and I hope that the authors will take them in serious consideration.

All the reviewers clearly agree (and so do I) on the quality of the system developed, and on its usefulness in the context of computer, and social, behavioural and economic sciences. This is why I firmly believe that it should be published in PeerJ CS at some point - even if not in the present form.

One of the main issues of the paper, according to all the reviewers, is that it has been structured as a system paper rather than a proper research journal paper. However, even if that is not completely clear in the guidelines for reviewers, PeerJ Computer Science actually allows the submission of "Application Note" papers (a.k.a. system papers), even if these kinds of papers should preferably present extensive test of the data presented and be accompanied by demonstrations of real-world applications.

Thus, the authors don't have to rewrite the paper to conform to a Research Article structure, but they should address the other relevant issues highlighted by the reviewers, and in particular:

- the lack of statistics about the current usage of the system;
- the lack of a proper evaluation and/or user testing session for the various tools presented.

Thus, all these aspects should be clearly addressed, extending the paper accordingly, so as to deserve a publication in PeerJ CS - that's why I've assigned "major revision" as final decision of the paper.

In addition, I would also like to see a full addressing of all the other relevant comments made by the reviewers, accompanied by an appropriate response letter.

·

Basic reporting

This is a report on the creation of the Observatory on Social Media, an infrastructure a) retrieving and storing Twitter data (tweets), and b) offering free analytical services build on these data. The long term goal is to extend this service to a diversity of social media data (not just Twitter) in order to serve the needs of the public and social, behavioral and economic (SBE) research.

Experimental design

There is no experimental design.

Validity of the findings

As far as I am aware, this is the first time that such an infrastructure is made available to the public, through services accessible through human and programmatic interfaces. This is a significant achievement, given the size of the dataset (70B tweets, with collection ongoing) and given that a number of services over this data are provided in free access. This is a real and unique benefit offered to the public.

Comments for the author

A couple of questions:
- I could not find policy guidelines on the usage of tweets / twitter users objects returned by the tools. Twitter mentions restrictions of 50,000 objects made available per day (https://dev.twitter.com/overview/terms/agreement-and-policy , section 6 b). The paper does not explain where OSoMe stands in regard to this?

- What is the relationship between OSoMe and Truthy? I understand that truthy is an ancestor to OSoMe, yet it seems to be more than that as the “botornot” service redirects to a truthy url. The footnote 1 in the paper does not explain the relationship clearly.

- Development plans. The description of the API says it offers access to “timelines, active users, counts, or tweet ids.” Yet the API doc indicates less than that, which suggests there is plan to expand the API in the future. The paper could share this roadmap.

Reviewer 2 ·

Basic reporting

- The article is well-written and clearly understandable. Figures are fitting the text.
- The motivation of why such an observatory is useful and how and by whom it can be used is clear and I fully agree that an easily accessible and searchable (by API or visual interfaces) Twitter data set that is constantly updated can be very valuable for CS and the “SBE” sciences – and that such a service is currently not available in this form. However, novelty is not a deciding PeerJ criterion.
- Regarding the PeerJ criteria, there are no clear research questions or hypotheses stated, rather, the software architecture and integrated Tools of OSoMe are described descriptively.
- While the website lists some related software code, none is referenced in the paper, particularly not the source code for the actual data collection system. There is no data being analyzed in the paper.

Experimental design

- there is no experimental evaluation conducted

Validity of the findings

- no findings are reported (see lacking experiments)

Comments for the author

##Fit to Aim & Scope of PeerJ##

- For me, this is not an “original primary research” paper as necessitated by the editorial criteria of PeerJ but what is sometimes called a “system paper”. There are no clear research questions or hypotheses stated, rather, the software architecture and integrated Tools of OSoMe are merely described descriptively. Most of the end-user tools have been previously proposed. There is not clear research question and no evaluation is carried out of any of the components of the system. I don’t see the research contribution of this paper; it’s the description of a platform. Even for a system paper, I would have expected some statistics on users, performance (e.g. how performant is the API, how many clients+requests does it handle), etc.
- The backend seems to be well-designed and employs some state of the art technology but also doesn’t really strike me as a new (research) contribution.

##Representativeness of the offered data##

- Given that research has pointed out (and is acknowledged by the authors) that collecting from Twitter’s APIs can introduce notable biases in what data is collected, I would have at least expected a discussion of this bias, coming from the Streaming API they use, as well the possible evolution of that bias over time, with changes implemented to Twitter’s technical infrastructure (the reported ratio change of geo-tagged Tweets might actually be an indicator for this, for instance).

- It’s also not clear what the authors refer to when talking about that “the data has been collected from a random 10% stream sample of public Twitter posts”: I might be mistaken, but is the streaming API not only giving 1% of all Tweets? Has this changed over the years? The whole Tweet collection process and what fraction of Tweets is stored in OSoMe is not transparent at all (And, as has been rightfully argued in e.g. Ruths & Pfeffer, 2014, size does *not* automatically heal bias; this holds especially when studying smaller subpopulations with this data).

- Optimally, readers and the users of OSoMe need to have an idea of what biases were captured with the collected data, as the resulting inferences drawn by users not familiar with the technical background of such a service, especially from the SBE sciences, can lead to grave misinterpretations. Not only is there no insight given into those possible biases or how they might have been addressed during data collection, the issue is basically not discussed at all for OSoMe.

- On top of the collected data, the tools used to visualize it add another layer of selectively filtering certain parts and structures of the data. There is no evaluation provided in this or the related publications for the tools that actually measures how these filters distort the representativeness of the data.
o Maps tool: only a very small percentage of tweets actually has geo-tags
o Network movies: the algorithm behind this tool selects only certain nodes and edges ("most relevant part of the network") which is not evaluated in terms of the meaningfulness of the results it produces.
o Network graph (static): only selects certain k-cores, for visualization purposes

- Given that OSoMe is supposed to be a service for end-users that have little to no knowledge of the data collection and subsequent processing, i.e. more or less blindly trust the representativeness of this data at least with respect to the ground population of all Tweets, I find it very troubling that these questions are not tackled.

##Usability and usefulness of end-user tools##

- Apart from any data issues, no evaluation is visible that shows that users (i) can operate, (ii) understand and appropriately interpret, and (iii) profit from using these tools. E.g., the network movies seems pretty confusing to me and I don't see how an average end-user can make much sense out of them. As the tools are advertised as a major part of this system, user studies are mandatory to see if they actually fulfill their goal.

- Minor: The loading times were in my own several try-outs quite high for an end-user-centric site and certainly higher then advertised in the paper (e.g. Maps tool)

Reviewer 3 ·

Basic reporting

The paper presents the IUNI Observatory on Social Media, which is an open platform for the analysis of large collections of tweets. This platform aims to facilitate data analysis by providing retrieval, visualisation and analytics tools.

The paper is well written, including a good introduction and well-referenced background. They describe their system architecture and introduce some of the system's tools. While I find this system highly relevant to the research community, I wonder if this is the correct medium for this paper. The paper is presented as a general report of the system rather than a system that has been thoroughly evaluated. At the moment it looks more like a technical report than a research paper. There is no evaluation on any of the presented tools nor as a user-based evaluation or in terms of the system’s time-performance for processing big data.

In that sense there is no hypothesis to be proved in the paper, which makes the paper short as a journal publication.

Experimental design

There is no experimental design to judge for this paper.

Validity of the findings

The paper presents a description of the set of tools provided in their system. However, no evaluation on the performance of these tools was presented.

Comments for the author

At the moment the paper is publishable as a technical report but not as a journal paper. I encourage the authors to gather the evaluations done on their system and describe such evaluations within this paper.

---

## Round 0.2 · accepted · Accept

· Academic Editor

Accept

First of all, I would like to thank the authors for having addressed all my comments and reviewers' suggestions in their revision, and thanks again to the reviewers for their wonderful job.

Since this is an Application Note, having the raw data available for reviewing is not a strict requirement for deserving publication. However I would like to see a clear statement in the camera ready version of the paper about data license (e.g. for the data retrieved by means of the API), which would clarify to what extend users can reuse such data for their purposes (research, commercial, etc.).

I would also like to ask authors to address reviewer's suggestion about the discussion of improvements/bottlenecks, by adding some paragraphs with a more detailed explanation.

There are also some typos that should be corrected in the text, e.g.:
- ." -> ".
- 2013b,a -> 2013a,b

Concluding, I think that the paper is now in a good shape for being published as Application Note, and I thus suggest acceptance.

Thanks again for submitting your great work at PeerJ CS.

·

Basic reporting

The authors provided adequate answers to my questions and remarks.

Experimental design

The authors provided adequate answers to my questions and remarks.

Validity of the findings

The authors provided adequate answers to my questions and remarks.

Comments for the author

The authors provided adequate answers to my questions and remarks.

Reviewer 2 ·

Basic reporting

As before; alas article is now an "Application Note" and I suppose most of the guidelines for research papers do not apply in that case. However, I did not had the appropriate guidelines for Application Notes available.

One major PeerJ requirement I'm not sure about if it applies here and which is for the Editor to decide is this:
"All appropriate raw data has been made available in accordance with our Data Sharing policy." and further
"Your reviewers must have your raw data or code to review. Please submit:
As a link to a repository where the data is accessible.
Uploaded as a supplemental file.
Generally however, there are very few circumstances in which we can accept a manuscript without raw data. It is not required if your submission does not deal with raw data or code.
Some examples of invalid reasons for not submitting raw data or code:
- The data is owned by a third-party who have not given permission to publish it within this article. Please add a note to PeerJ staff to not publish the raw data alongside your article.
- The raw data is too large. Please upload the raw data to an online repository (e.g. Figshare, Dryad etc)."
--> If that is a requirement for Application Notes as well, this paper would not qualify as neither the source code of the system nor the data about the performance evaluation is available. (the twitter raw data is of course not possible to include)

Experimental design

-

Validity of the findings

-

Comments for the author

Thanks to the authors for addressing most of my concerns and questions.

- As for my remarks regarding research questions, testing etc. required in a research article, these are basically nullified by this now being submitted as an Application Note, which I take to be the same as what is commonly referred to as a Systems Paper.

- Regarding the bias in the data I appreciate the very clear and honest disclaimer that was added in the "Data Source" section. This caveat of the system should also be made clear in any documentation and on the website for prospective (SBE) users.

- The evaluation gives some insight into the performance and future avenues for the architecture. I would've actually liked to see an even deeper discussion of the possible architecture improvements and current bottlenecks, since this is now a system paper/ app. note.

- There is no evaluation added on the usability/usefulness of the visualization tools, which is beyond the scope of an application note, as far as I can judge. However, I would appreciate a better outline of *how* the authors intend to evaluate these tools, not just a promise that they will.